# Molecular Toxicology and Pathophysiology of Comorbid Alcohol Use Disorder and Post-Traumatic Stress Disorder Associated with Traumatic Brain Injury

**DOI:** 10.3390/ijms24108805

**Published:** 2023-05-15

**Authors:** Zufeng Wang, Chengliang Luo, Edward W. Zhou, Aaron F. Sandhu, Xiaojing Yuan, George E. Williams, Jialu Cheng, Bharati Sinha, Mohammed Akbar, Pallab Bhattacharya, Shuanhu Zhou, Byoung-Joon Song, Xin Wang

**Affiliations:** 1Department of Neurosurgery, Brigham and Women’s Hospital, Harvard Medical School, Boston, MA 02115, USA; 2Department of Forensic Medicine, Soochow University, Suzhou 215006, China; 3Division of Neuroscience & Behavior, National Institute on Alcohol Abuse and Alcoholism, National Institutes of Health, Rockville, MD 20892, USA; 4Department of Pharmacology and Toxicology, National Institute of Pharmaceutical Education and Research (NIPER), Ahmedabad, Gandhinagar 382355, Gujarat, India; 5Harvard Medical School, Harvard Stem Cell Institute, Boston, MA 02115, USA; 6Section of Molecular Pharmacology and Toxicology, National Institute on Alcohol Abuse and Alcoholism, National Institutes of Health, Rockville, MD 20892, USA

**Keywords:** alcohol use disorder, post-traumatic stress disorder, traumatic brain injury, comorbidity, characteristics, molecular toxicology, metabolomics, inflammation, neuroendocrine, signal transduction pathways, genetic and epigenetic regulation

## Abstract

The increasing comorbidity of alcohol use disorder (AUD) and post-traumatic stress disorder (PTSD) associated with traumatic brain injury (TBI) is a serious medical, economic, and social issue. However, the molecular toxicology and pathophysiological mechanisms of comorbid AUD and PTSD are not well understood and the identification of the comorbidity state markers is significantly challenging. This review summarizes the main characteristics of comorbidity between AUD and PTSD (AUD/PTSD) and highlights the significance of a comprehensive understanding of the molecular toxicology and pathophysiological mechanisms of AUD/PTSD, particularly following TBI, with a focus on the role of metabolomics, inflammation, neuroendocrine, signal transduction pathways, and genetic regulation. Instead of a separate disease state, a comprehensive examination of comorbid AUD and PTSD is emphasized by considering additive and synergistic interactions between the two diseases. Finally, we propose several hypotheses of molecular mechanisms for AUD/PTSD and discuss potential future research directions that may provide new insights and translational application opportunities.

## 1. Introduction

In recent years, the unsettling prevalence of a feature in most veterans with a history of traumatic brain injury (TBI) has been that a high proportion of post-war veterans diagnosed with a psychiatric problem so-called post-traumatic stress disorder (PTSD) also suffer from alcohol use disorder (AUD) [1,2,3,4]. Compared with patients with either AUD or PTSD alone [5,6], the clinical manifestations of the co-occurrence of AUD and PTSD (AUD/PTSD) are more serious and the prognosis is more devastating [7,8]. According to numerous survey reports, veterans who were diagnosed with AUD/PTSD had higher rates of clinical depression, impaired executive function, generalized anxiety disorder with poor sleep, suicidal ideation and behavior, and other mental or neuropsychiatric disorders than veterans who were not diagnosed with AUD/PTSD [1,9]. Besides its prevalence in veterans, AUD/PTSD is also seen in other clinical conditions, resulting in increased severity of those disease conditions [10,11]. Many adverse manifestations of comorbid AUD and PTSD are known, yet the mutual relationship and additive and synergistic mechanisms are poorly understood.

AUD is a chronic relapsing brain disorder characterized by an impaired ability to cease or regulate chronic and excessive alcohol misuse despite adverse social, occupational, or health consequences. The 12-month prevalence and lifetime prevalence of AUD are 13.9% and 29.1%, respectively [12]. The characteristics of AUD include a high prevalence of concurrent long-term misuse and dependency with withdrawal symptoms [12,13]. Unfortunately, AUD individuals are more likely to be involved in accidents, slips, and falls to incur brain trauma, with increased chances of developing PTSD after TBI compared with those without AUD [14]. Patients with PTSD are more likely to misuse alcohol. Epidemiological studies on post-war veterans have shown that those with PTSD are likely to increase AUD by two-fold compared with those without PTSD [15,16]; AUD and PTSD interact to accelerate the disease’s progression and produce more severe AUD/PTSD symptoms [17,18,19]. 

TBI, resulting from an impact by an external mechanical force, is the leading cause of mortality and disability across all ages, both in civilian and military populations [20]. The degree of brain damage divides TBI into three categories: mild, moderate, and severe. Among these, mild TBI (mTBI) accounts for the largest proportion of TBI. The type of TBI is a critical factor for the patient’s overall future prognosis and well-being [21,22]. Previous studies show that between 30% and 50% of TBI patients are alcohol intoxicated when they are physically hurt [23,24]. Epidemiological studies have shown that mental health problems with depression commonly occur after TBI, especially mild TBI (mTBI), and a significant proportion of TBI individuals develop PTSD [25,26]. About 10% of persons exposed to a potentially life-threatening traumatic event have a chance of acquiring PTSD [27]. Other epidemiological data show that PTSD prevalence in the lifetime, past 12-months, and past 6-months is 8.3%, 4.7%, and 3.8% respectively [28,29]. About one-third of patients with PTSD struggle with the misuse of various habitual substances, while AUD is the most prevalent substance use disorder (24.1%) among individuals with PTSD [30]. PTSD is the result of multiple damaging factors [3,7]. Among these factors, TBI is an important risk factor for the occurrence of PTSD [31]. TBI is a highly heterogeneous injury state resulting in a patient population with markedly different injuries, comorbidities, and predicted outcomes. The long-term sequelae of TBI can include psychiatric and neurological dysfunction, as well as a whole host of non-neurological diseases. TBI, PTSD, and AUD interact with each other and have complex relationships. One focus of our review is on the comorbid AUD and PTSD following TBI. Alcohol misuse is one of the strongest predictors of TBI, and a significant proportion of TBI patients develop a problem of alcohol misuse [23]. Compared with non-drinkers, drinkers have about a four-fold higher incidence rate of TBI in their lifetime [32]. A large retrospective review reported that PTSD was associated with a three-fold increase in the risk of developing AUD [28]. 

Not all TBI victims suffer from PTSD, AUD, or PTSD, particularly following TBI, and they have been documented not only in military veterans but also in community patients [25]. However, there is a knowledge gap in our understanding of the molecular toxicological and pathophysiological mechanisms and behavioral consequences of AUD/PTSD. Most people, including healthcare professionals, only comprehend AUD/PTSD as a combination of the two diseases without understanding the more severe medical and socioeconomic implications of this combination, which may have more negative consequences for affected individuals and society [33]. It is difficult to objectively identify the state markers of AUD or PTSD [34,35]. As a result, there is currently an urgency in understanding the reason why or under what conditions these patterns emerge and progress, as well as the molecular toxicology and pathophysiological mechanisms underlying the comorbidity. These problems and consequences must be addressed and studied to correctly diagnose and treat patients who exhibit AUD/PTSD-associated neurobehavioral symptoms.

## 2. Materials and Methods

This review was carried out by searching the keywords (AUD, PTSD, and TBI) for the topics “Molecular Toxicology” or “Pathophysiology” using MESH terms. The literature searches were performed by using the PubMed/Medline databases. Only English-language articles were chosen. We examined the titles and abstracts of articles and then obtained the full texts of potentially relevant papers. 

## 3. The Main Characteristics of AUD/PTSD

Alcohol (ethanol) is the most frequently used recreational agent [36] and plays a significant role in many societies and cultures worldwide [37]. The diagnostic criteria for AUD have also evolved with each iteration of the *Diagnostic and Statistical Manual of Mental Disorders* (DSM). Under DSM-5, anyone meeting any 2 of the 11 criteria during the same 12-month period receives a diagnosis of AUD; the severity of AUD (mild, moderate, or severe) is determined by the number of criteria met [38,39,40] (Table 1). PTSD is characterized by four symptom clusters (Table 1): constant re-experiencing, avoidance of stimuli associated with the traumatic event, increased physiological arousal, and negative alterations in mood and cognition, thus resulting in significant impairments in daily activities [41]. For instance, many patients with PTSD specifically display the following behavioral changes: avoidance of stimuli connected to the trauma; recurring, involuntary, and intrusive recollections; negative cognitive and mood changes; chronic or episodic anxiety with sleep difficulty [41]. 

Clinician and other medical professionals generally regard AUD/PTSD as a combination of neuropsychiatric symptoms. However, more and more research data show that the comorbidity of AUD and PTSD is not just a simple combination of the two disorders; it is more serious and harmful than the simple addition of the two disease conditions, demonstrating a synergistic relationship with a feed-forward vicious circle [42,43]. Patients suffering from AUD/PTSD following TBI show not only typical symptoms of individual AUD and PTSD but also more severe mixed symptoms such as pain, severe PTSD symptom clusters, mental disorders, neurocognitive impairment, poor overall nutrition, alcohol-related gut leakiness with elevated endotoxemia, local or systemic inflammation, cardiomyopathy, pancreatitis, liver fibrosis, cirrhosis, and several types of cancer [17,18,19,44,45]. In short, the symptoms of AUD and PTSD comorbidity represent significant synergistic effects with more severe and aggravating individual symptoms. Patients’ severe physical and mental symptoms cause a significant reduction in their various social activities and a lack of enthusiasm for seeking treatment or eating a balanced diet eventually leads to social issues along with the disease’s progression [33]. As a result, the harm caused by the comorbidity of AUD and PTSD is not only a medical issue but also has evolved into an economic and social problem that is gaining attention.

We summarize the main characteristics of comorbidity between PTSD and AUD associated TBI (Table 1): (1) significant additivity of clinical symptoms with multiple indications of the two diseases, (2) severe hazards to produce more serious clinical manifestations than any single disease in a vicious circle, (3) synergistic effect on the interaction and pathogenesis of AUD/PTSD, and (4) with very little objective identification of state markers, the effectiveness of diagnosis and treatment is unsatisfactory. However, due to a lack of research in this area, very little is known about the molecular toxicology and pathophysiological mechanisms of AUD/PTSD. Therefore, a thorough investigation of the toxicological and pathophysiological mechanisms of AUD/PTSD is required, while a precise study on the additive or synergistic interactions between AUD and PTSD for improving treatment efficacy and the objective identification of state markers is needed.

## 4. Molecular Toxicology and Pathophysiological Mechanisms of AUD/PTSD 

Although some research suggests that low levels of drinking can exhibit some health benefits, AUD has been widely identified as a major risk factor for both mental and physical harm and disability [46,47]. PTSD is a debilitating condition that can commonly develop after TBI and increasing evidence indicates a possible link between PTSD symptoms and various substance use disorders [48]. Higher levels of post-treatment dysfunction and long-term alcohol misuse problems are associated with poorer responses to PTSD treatment, implying a link between AUD and PTSD [49]. The understanding of the synergistic interactions between AUD and PTSD, as well as their comorbid molecular toxicology and pathophysiological mechanisms have become critical for better prevention and treatment of AUD/PTSD symptoms. 

AUD can cause neurochemical and behavioral changes in the brain. TBI disrupts the normal physiological state of the brain, resulting in changes in brain structure and metabolism, so patients with PTSD following TBI may have a series of alterations in brain molecular components. Despite significant advancements in understanding the molecular complexity of AUD and the dynamics of the action mechanism of alcohol misuse [50,51,52], identifications of objective state markers that can be universally recognized are needed for AUD [53]. The objective identification of state markers and effective treatment for PTSD is more challenging [54,55]. The brains of patients with AUD/PTSD show complex signs of mixed changes in the molecular components of the two diseases. Table 2 summarizes some of the molecular and biochemical changes, including metabolism/pharmacokinetics, inflammatory factors, and neuroendocrine alterations, that occur in the brain as a result of PTSD, AUD, and AUD/PTSD.

The following sections summarize the toxicological and pathophysiological mechanisms of PTSD, AUD, and AUD/PTSD based on the aspects of metabolomics, inflammation, neuroendocrine, signal transduction pathways, and genetic and epigenetic regulation. 

### 4.1. Metabolomics in AUD/PTSD

The normal brain’s activities and responses are regulated by the need to maintain proper homeostasis in the presence of various endogenous and exogenous challenges. Following a TBI, maintaining optimal metabolic function in the brain is critical for preventing the onset and progression of neurological and psychiatric disorders. Targeted metabolomic quantitative analysis of blood and urine specimens is a common research methodology [71,72]. The metabolic characteristics of alcohol dependence or misuse can be determined by metabolomics and machine learning tools [73]. Alcohol misuse not only affects the redox reactions, with the production of acetaldehyde and acetic acid in the body but also decreases the production and utilization of energy in the brain and other organs. Alcohol cytotoxicity is frequently observed in patients with long-term alcohol misuse. 

The metabolomic analysis of the post-mortem human brain tissues revealed that chronic excessive drinking may have a preferential effect on catecholaminergic signals [74]. Through metabonomic analysis of the changes in the body caused by AUD or alcohol dependence, some trait or state markers are found; these provide very important objective biomarkers for forensic and clinical toxicology [75,76]. For example, phosphatidylethanol is a promising marker for monitoring alcohol use in pregnancy [77], while salivary exoglycosidases and deglycosylation processes can be biochemical markers for chronic alcohol consumption and dependence [78].

PTSD is known to be associated with metabolic dysfunction, which may be caused by changes in the composition and abundance of gut microbiota [79]. Metabolic disorders in PTSD veterans include increased glycolysis flux to lactate, a decreased tricarboxylic acid cycle, and impairments in amino acid and lipid metabolism [80]. Akbar and colleagues reported that PTSD in adolescents can lead to increased mitochondrial dysfunction and neuronal cell death [81]. In a rat experiment, catalase activity in the cortex increased after TBI, peaking at 3× on the third post-trauma day [56]; an elevated production of acetaldehyde in the brain and salsolinol (a combined product of acetaldehyde and dopamine) increased voluntary ethanol intake in rats [82]. According to these findings, PTSD following TBI can increase the activity of brain catalase, which could lead to increased production of acetaldehyde following alcohol consumption; elevated acetaldehyde further stimulates voluntary alcohol consumption, resulting in chronic alcohol misuse.

Catalase and cytochrome P450-2E1 (CYP2E1) are the two major enzymes that are responsible for brain ethanol oxidation into acetaldehyde [57,83] (Table 2). In addition, mitochondrial aldehyde dehydrogenase 2 (ALDH2) may be a significant therapeutic target against alcohol-induced organ and tissue damage [84]. Elevated CYP2E1 has been reported to be linked to increased oxidative stress and cell death signaling in a variety of tissues [57,84]. Because these brain regions are associated with memory, potentiation, autonomic functions, and dopamine regulation, CYP2E1 is likely to play a key role in increasing ethanol intake after TBI via acetaldehyde production [85,86,87,88], while increased CYP2E1-related oxidative stress may contribute to alcohol-induced tissue injury, including the brain [57]. In addition, inhibiting CYP2E1 has been shown to increase dopamine concentration in the brain, implying that more research is needed to clearly define the role of CYP2E1 in AUD-related toxicities and neurobehavioral changes [86].

Brain acetaldehyde has been found to have a reinforcing role in voluntary ethanol consumption, though the mechanism remains controversial [88,89,90,91,92,93]. Since acetaldehyde cannot pass the blood-brain barrier (BBB) while ethanol can, acetaldehyde in the central nervous system (CNS) can only be produced from ethanol oxidation [93,94]. However, because catalase is a key antioxidant defense enzyme, a decrease in catalase activity in the brain has been linked to an increase in oxidative stress, which may contribute to cellular damage and cognitive impairments [95,96]. A recent study showed that ALDH2 deficiency, the major enzyme responsible for metabolizing acetaldehyde, is a risk factor for alcohol-mediated gut and liver injury in rodents and humans [84]. 

Accumulating evidence suggests that glutamate neurotransmission plays a critical role in alcohol dependence. A recent study reported that glutamate is modulated by alcohol cues in the forebrain reward circuit pathway in the early stage of alcohol dependence [58]. In the initial stages of TBI, extracellular glutamate levels increase and glutamate buffering and clearance are impaired. The coordination of glutamate release and uptake is critical to regulating synaptic strength, long-term potentiation, depression, and cognitive processes in chronic TBI [97]. A large body of preclinical work supports the targeting of pre- and post-synaptic glutamate signaling to ameliorate PTSD [98]. Rat studies have also revealed that AUD can cause several abnormalities in the gene expression of the proteins responsible for cholesterol metabolism, leading to long-term cholesterol metabolism disorders [59]. Phosphatidylethanol is a metabolite of non-oxidative ethanol metabolism and this molecule can be used as a biomarker for alcohol consumption [60] (Table 2). According to a recent study, the alpha6-nicotinic acetylcholine receptor (a highly sensitive target of alcohol consumption) mediates low-dose ethanol effects via a positive allosteric mechanism [61] (Table 2). 

### 4.2. Changes in Inflammatory Cytokines 

Due to the toxic effects of alcohol, especially in large amounts, the brain and other organs frequently show severe functional and structural abnormalities in AUD patients and experimental models. Additionally, the consumption of ethanol has been linked to neuronal inflammation through microglial activation, neuritis, and higher levels of monocyte chemotactic protein 1 (MCP-1) mRNA and protein expression [14,63]. However, the effect of alcohol consumption on the increased pro-inflammatory cytokines and prognosis of patients with brain injury is still up for debate. According to some research findings, TBI with a positive blood alcohol concentration (BAC) is associated with lower systemic Interleukin-6 (IL-6) levels and white blood cell counts, indicating that ethanol may cause an immunosuppressive response [64] (Table 2). Another study with RNA sequencing analysis found a link between alcohol consumption and miRNA expression; functional enrichment of target genes in miRNA-related families (especially the miR-183 and miR-200a/b families) suggests that Toll-Like Receptor 4 (TLR4) signal transduction is linked to the neural inflammation pathway network in an AUD mouse model [99].

Pro-inflammatory cytokines, which play an important role in PTSD, are produced after TBI and participate in the upregulation of the inflammatory response. For instance, TBI primarily increases the production of pro-inflammatory cytokines, which are involved in the upregulation of inflammatory reactions and play an important role in maintaining normal brain function and repair after TBI [100,101]. In addition, patients with PTSD are more likely to have significantly elevated levels of C-Reactive Protein (CRP), which is a commonly used marker for monitoring peripheral inflammation [62,102]. Immune dysfunction and abnormal expression of pro-inflammatory and/or anti-inflammatory cytokines are frequently associated with PTSD. The increased expression of interleukins and pro-inflammatory cytokines after PTSD may also play an important role in the pathophysiology of PTSD [103]. 

### 4.3. Neuroendocrine Alterations 

Over the last decade, studies have demonstrated that the dysregulation of the endogenous stress-regulation systems plays a crucial role in a variety of psychological disorders, including PTSD [104] and AUD [105,106]. PTSD is a neuropsychiatric disorder. Stress-related disorders (such as PTSD, depression, panic disorder, and chronic anxiety) have long-term consequences in the neuroendocrine system. The hypothalamic–pituitary–adrenal (HPA) axis is primarily responsible for stress neuroendocrine regulation and post-stress homeostasis recovery after stressful events [107]. PTSD following TBI has been shown to cause a wide range of neuroendocrine changes in the brain, potentially leading to the development of AUD. In fact, several studies have found that exposure to traumatic memories increases the desire to drink alcohol, leading to AUD/PTSD [108]. 5-HT (5-HydroxyTryptamine, also known as Serotonin) receptors are closely related to increased alcohol consumption. Ethanol can rapidly activate 5-HT3 receptors, resulting in rapid excitatory 5-HT transmission and increased synaptic 5-HT coupled with 5-HT3 receptor responsiveness. As a result, the increased reactivity of synaptic 5-HT and 5-HT3 receptors can increase dopamine transmission and flow in the reward pathway [65]. A recent genetic association study reported that the genotype of the serotonin transporter-linked polymorphic region (5-HTTLPR) is the key mediator of the interaction between stress and PTSD symptoms [66] (Table 2). 

It is well-established that the major neurotransmitter dopamine is involved with reward systems, motivation, euphoria, fear, fright, anxiety, and learning. According to a recent study, the physiological properties of dopaminergic neurons can predict an individual’s susceptibility to stress and anxiety [109]. In addition, the dopamine D3 receptor (DRD3) and the brain-derived neurotrophic factor (BDNF) were found to be linked to alcohol consumption. A recent study discovered a link between haplotypes in the BDNF gene and AUD [67]. A study of both pre- and post-deployment military service members found that those who had an mTBI prior to their combat-related traumatic stress had higher levels of a specific BDNF genotype than those who did not have an mTBI. Soldiers who sustained an mTBI during their service were found to be more likely to suffer from traumatic stress than those who did not [68]. Thus, monitoring circulating growth factors in the blood, such as BDNF, will help in stratifying AUD patients at various stages of the alcohol withdrawal treatment plan and in predicting treatment efficacy [69] (Table 2). 

In recent years, a research direction in the study of AUD that has received increased attention is the “gut-brain axis” (the bidirectional interactions between the brain, gut, and microbiome) and its potentially huge influence not only on physical and digestive issues but also on mental health conditions [110]. Studies revealed the close interrelationships between HPA, gut-associated immune tissues, and the enteric nervous system under stress. The steroid hormone cortisol is also known as hydrocortisone when administered as a medicine. Compared with persons without irritable bowel syndrome (IBD), patients with IBD have a higher cortisol/dehydroepiandrosterone ratio after awakening, and the elevated ratio peaks under continuous stress [110]. Studies mostly based on the measurements of saliva or plasma of chronic drug users or binge/disordered alcohol drinkers [111] show that baseline cortisol levels in chronic substance users significantly increase relative to controls [112]. Alcohol consumption activates the HPA axis and causes it to release cortisol/corticosterone. Clinical investigations and laboratory testing have linked the disturbance of the HPA axis to AUD [113,114]. Chronic alcohol exposure enhances the HPA axis’s tolerance to alcohol, resulting in lower blood cortisol levels after alcohol consumption in AUD patients compared with individuals who do not have AUD [115]. Because individual cortisol assessments are strongly influenced by the acute context of the measurement conditions (e.g., time of day, day of the week, specific circumstances, assay methods, etc.), short-term cortisol measurement cannot reflect the secretion status of long-term cortisol secretion. On the contrary, hair cortisol concentration (HCC) analysis [116,117] offers the most reliable method for measuring the long-term activity of the HPA axis through a retrospective assessment of cumulative cortisol exposure over time for both basic research and clinical purposes [118]. Research demonstrates a close relationship between long-term alcohol consumption and HPA axis activity; elevated HCC in chronic substance users of alcohol, psychostimulants, and opioids have also provided a retrospective index of cumulative cortisol exposure over periods [119,120]. 

Increasing research evidence supports those gut-brain peptides, such as glucagon-like peptide 1 (GLP1), ghrelin, neuropeptide Y, substance P, and neurotensin, that are thought to play an important role in the molecular toxicology and pathophysiology of AUD-associated tissue damage [70,121,122,123,124,125] (Table 2). Changes in microbiome composition and abundance in the intestine during stress exposure are related to lower-gut microbial diversity in PTSD [126,127]. Targeting the gut microbiota with fecal microbial transfer (FMT) from healthy donors or taking probiotics may help patients with AUD/PTSD through changes in their gut-brain axis. Although this research strategy has been used in the study of other diseases [128,129], this translational application will necessitate more large-scale randomized clinical trials in the future.

### 4.4. Altered Signaling Transduction Pathways in the Brain 

Alcohol can cause glial activation, neuronal apoptosis with myelin loss, and eventually long-term neurological deficits via oxidative stress signals; abnormal cell signaling transduction is crucial in alcohol-induced brain injury [130]. Some studies have suggested that multiple signal transduction pathways, such as mammalian target of rapamycin (mTOR), nuclear factor kappa-B (NF-κB), Mixed Lineage Kinase Domain-Like (MLKL), etc., play a role in alcohol-induced brain injury [131,132,133].

Previous research has shown that after a TBI, the brain experiences a significant change in functional connectivity and transduction, disrupting a wide range of brain circuits [134,135]. Anatomical and structural variations in the signaling pathways may be influenced by genetic and environmental factors, leading to functional and behavioral changes that may predispose to the clinical manifestations of PTSD [136]. Numerous studies have confirmed that the trauma factors of PTSD following TBI can cause damage to the BDNF-Tyrosine Kinase receptor B (TrkB) signaling pathway. Increased levels of BDNF and TrkB, as well as epigenetic regulation of BDNF genes, play an important role in the long-term fear memory of PTSD patients [137]. PTSD can activate the transcription factor Cyclic Adenosine monophosphate (cAMP) response element binding protein (CREB) and disrupt the CREB-BDNF signaling pathway, which can lead to a range of clinical symptoms [138].

AUD can reduce the number of synapses and cause axonal impairment in the medulla oblongata, making the brain more vulnerable to further injury. High mortality rates have been statistically linked to synaptic and axonal disruptions caused by comorbid AUD and PTSD [139]. A growing body of evidence suggests the existence of molecular mechanisms that maintain alcohol consumption [140]. This provides a new opportunity to create a novel treatment or preventive strategy for AUD-associated neurotoxicity by regulating intracellular signaling pathways. A recent study found a link between the pathophysiology of AUD and an abnormality of N-methyl-d-aspartate receptor (NMDAR) signaling in the nucleus accumbens (NAc) [141]. More research is needed to investigate the neurotoxic pathogenesis of AUD and AUD/PTSD.

### 4.5. Genetic and Epigenetic Changes in the Brain

Many studies have shown that AUD can cause extensive changes in gene expression and epigenetic regulation. In fact, long-term alcohol misuse has been linked to changes in DNA modifications and gene expression, as well as alterations in functional characteristics similar to those seen in AUD patients with abnormal behavior and decreased cognitive capacity [142]. Differentially inherited genetic and epigenetic alterations may explain distinct biological responses to traumatic exposure when facing the same risk [27]. Table 3 summarizes the genetic changes in the brain for AUD, PTSD, and AUD/PTSD.

According to the study of alcohol-related gene variation and brain function change, the severity of AUD-related consequences seems to be related to the altered responses of the frontal lobe of the brain to ethanol signals; the genetic variation of dissimilar neural plasticity and signal pathways is the main reason for this different response [138]. Genome-wide sequencing analysis reveals that DNA methylation may affect the gene expression pattern in AUD and PTSD symptoms and may play a role in their pathogenesis and progression [143,154]. Evidence suggests that AUD has polygenic risk factors and is associated with neurological changes. As a result, more research is needed on the relationship between the sum of genetic or epigenetic changes and alterations in brain function (including behavior and cognitive ability). 

Differential gene expression and epigenetic changes are known to be controlled by a complex set of regulatory systems. Negative changes in genomic DNA and epigenetic regulation can cause a variety of CNS disabilities. A genome-wide association study revealed that PTSD is found to be strongly linked with a single nucleotide polymorphism (SNP) in the coding region of CRP [62]. Furthermore, the study of gene polymorphism in veterans showed that when the vulnerability factors, such as BDNF, are consistent with the incidence of mTBI, changes similar to neural tissue degeneration in the brain can be induced [134]. These studies provide new insights into genetic and epigenetic mechanisms in the toxicology and pathophysiology of PTSD.

CYP2E1 is a major ethanol-inducible enzyme and is involved in alcohol-induced oxidative tissue injury [57]. Chronic alcohol exposure increases CYP2E1 protein expression in the hippocampus, cerebellum, and brain stem, and its elevation correlates with increased ROS levels [155]. Interestingly, in a rat model of TBI, scientists used real-time polymerase chain reaction to discover that CYP2E1 mRNA is elevated in the hippocampus [144]. Another study shows that region-specific CYP2E1 elevation after TBI increases acetaldehyde production in the hippocampus, medulla, and substantia nigra of the brain [156]. Furthermore, increasing ethanol consumption activates CYP2E1, which promotes the gradual development of toxicities associated with AUD [144,157,158,159] (Table 3). 

Numerous researchers recognize the theory that alcohol exposure can regulate epigenetics as a factor in the development and progression of AUD [160]. Meta-analysis and genome-wide significance studies have yielded a wealth of useful information. Polygenic risk factors, genetic correlations, cell-type group partitioning, and heritability enrichment analyses are found to be useful in analyzing the risks associated with alcohol consumption [161]. Another review by Zhang and Gelernter finds that long-term drinking can alter DNA methylation, induce epigenetic changes, and lead to neuroadaptation, which is linked to the mechanism and risk of alcohol dependence or AUD [143]. Genome-wide significance studies show that emotional memory formation decreases as the number of rs3852144 G-alleles increases, making individuals more resistant to the development of PTSD [150].

Long noncoding RNAs (lncRNAs) play an important role in brain development and synaptic plasticity. The BDNF-AS lncRNA regulates synaptic plasticity in the amygdala of AUD patients who begin drinking in adolescence via epigenetic reprogramming [146]. Ethanol exposure causes changes in miRNA expression in animal or cell culture models. The downregulation of miR-130a may result in changes in the expression of several genes in the prefrontal cortex (PFC) of AUD individuals’ postmortem brains [147]. Animal experiments show that the downregulation of MiR-142 reduces neuroinflammatory mediators and increases synaptic protein expression in the hippocampus, thereby improving PTSD-like behavior [148]. Boutte et al. find that copy number variation, especially the variation at chromosome 11 q14.2, can affect brain structural change and potentially influence AUD behavior [149] (Table 3). Alcohol consumption is linked to acute increases in cerebral blood flow (CBF), particularly in the frontal regions, which are associated with the risk of future heavy drinking and alcohol-related problems [162].

At the transcriptional level, ethanol inhibits the upregulation of a subset of immediate–early genes encoding for transcription factors, including Atf3, c-Fos, FosB, Egr1, Egr3, and Npas4, but does not affect the upregulation of others (e.g., Gadd45b and Gadd45c) [151] (Table 3). Notably, Chandrasekar et al. report that in a mouse model, except for c-Fos, Egr1, and Dusp5, most genes are sensitive to ethanol only when administered prior to TBI but not afterward [151]. Recent blood transcriptome research suggests that GRIN3B’s mRNA level might serve as a good potential early biomarker for the onset and progression of PTSD [152]. A recent epidemiological study shows the relationship between parental and offspring AUD individuals (with or without other mental disorders) and finds that offspring with parental AUD are at increased risk of AUD, regardless of parental mental disorder exposure [153]. Many researchers recognize that changes in genes and epigenetics in the brain play an important role in AUD, PTSD, and their comorbidity [27,163,164,165,166]. However, there is no clear conclusion on the mechanism by which AUD, PTSD, and comorbidity play a role in changing gene expression or affecting epigenetics; thus, this area requires additional research.

Emerging from the review, we summarize some interesting data in Table 4. It includes the most relevant data from epidemiology, metabolism, inflammatory factors, neuroendocrine components, and genetic studies.

## 5. A Variety of Possible Hypotheses Leading to AUD/PTSD

Based on the above discussion about the molecular toxicology and pathophysiological mechanisms that may lead to AUD/PTSD, we further propose the following hypotheses for the onset and progression of AUD/PTSD.

### 5.1. Metabolic Dysfunction Hypothesis

Due to the prolonged stressed state resulting from TBI, normal physiological regulating mechanisms in the brain may be disrupted and metabolic dysfunction with neuroendocrine abnormalities may appear in AUD/PTSD. Furthermore, an aberrant buildup of several metabolic intermediates may raise the chance of developing comorbid AUD and PTSD. Consequently, metabolic dysfunction is likely to play an important role in the development and progression of AUD/PTSD.

### 5.2. Inflammation Hypothesis

Biochemical changes in the body, as well as the activation of microglia and astrocytes, all contribute to the major inflammatory responses in the brain of AUD/PTSD. Consequently, a slew of pro-inflammatory cytokines and chemokines are activated and released into the extracellular space. This chronic inflammatory status causes structural and functional changes in the brain. In this regard, pro-inflammatory cytokine levels could be used as biomarkers for quantifying AUD/PTSD at an early stage of progression. As a result, an exaggerated inflammatory response may be a critical factor in the onset and duration of AUD/PTSD.

### 5.3. Neuroendocrine Alterations Hypothesis 

Neuroendocrine homeostasis is the foundation of the body’s normal functions. Under stimulation factors such as alcohol misuse, TBI, and PTSD following TBI, the neuroendocrine homeostasis, particularly in the nervous system, is disrupted and the levels of various neurotransmitters and their receptors are altered. 

The stimulation factors continue to exist and result in the continuation and development of AUD/PTSD, leading to the formation of a vicious circle. Neuroendocrine changes are not only an important cause of the onset of AUD/PTSD but also a key factor in its persistence through the formation of a vicious cycle. The investigation of neuroendocrine dysfunction in vivo is critical in determining the persistent states of AUD, PTD, and AUD/PTSD. 

### 5.4. Genetic Changes Hypotheses

AUD and PTSD can cause changes in brain homeostasis, which can alter gene expression and/or epigenetic regulation. Patients and experimental rodents with altered expression of various genes governing the sensitivity within cerebral neurons may eventually exhibit a variety of abnormal behaviors and decreased cognitive functions. However, the conditions and mechanisms regulating gene expression in vivo remain unknown; additional research is needed to fully understand their precise mechanisms. Thus, genetic and epigenetic changes may play a crucial role in the final development of AUD/PTSD, which is also an important reason why this comorbid disease is difficult to treat.

### 5.5. Integrated Hypotheses 

Based on our summary of the various mechanisms, we are currently considering the following scenario. Although there are various hypotheses for the molecular mechanisms of comorbid AUD and PTSD, none of them can perfectly explain its pathogenesis. Therefore, we consider that AUD/PTSD may be the result of the combined action of multiple pathogenic factors. The destruction of anatomical structure and the changes in neurotransmitters and their receptors may lead to alterations in metabolism, neuroendocrine function, pro-inflammatory factors, and others in the body. At the same time, the abnormality of the signal transmission system and neural function in the body caused by mental stress may lead to genetic and epigenetic changes. Finally, the aforementioned factors caused by the psychiatric disorder PTSD and/or AUD interact to cause AUD and PTSD comorbidity. Thus, comorbid AUD/PTSD results from the combination of several complex risk factors (Figure 1). 

## 6. Implications and Further Research Directions 

Because the causal relationship and mutual influence among AUD, PTSD, and AUD/PTSD are complex, many theories such as the self-medication hypothesis have been proposed to understand their relationship [167]. For instance, the decision of the individual to use a particular substance is influenced to a certain extent by the effect of the substance on subjective painful or unpleasant emotional states [168]. Although there are some research results, further detailed studies and discussions are needed. It is very important for clinicians to realize the complexity of AUD/PTSD based on their co-existence and the emergence of a more serious condition. With poor personal management and high suicide rates [169] among AUD/PTSD patients, integrated care and management by the multidisciplinary team should be highly encouraged. 

A lack of understanding of the disease’s molecular pathogenesis results in improper prevention and intervention in current treatments. This is also true in the case of comorbid AUD/PTSD. Therefore, we suggest that future studies should focus on the following aspects. (1) Molecular mechanisms of the enzymes such as catalase, CYP2E1, and ALDH2 are involved in the cerebral ethanol metabolism and oxidative stress, leading to brain injury before and during AUD/PTSD. (2) The HPA axis is principally in charge of stress neuroendocrine control and the recovery of post-stress homeostasis following stressful events. Studies on psychiatric patients diagnosed with more than a single mental disorder or on substance users with psychiatric disorders have shown that the co-occurrence of psychiatric disorders might be an additional correlate of HPA-axis dysregulation [170,171]. Regulation of neuroendocrine molecules such as serotonin, dopamine, and BDNF and signal transduction pathways such as BDNF-TrkB, CREB-BDNF, and NMDAR-related routes in the brain following AUD/PTSD events. (3) Examination of synaptic transmission and plasticity changes in AUD/PTSD events should be one of the focuses of future studies. Evidence from multiple sources has indicated that synaptogenesis formation relies on the precise signaling of neurotransmitters [25]. (4) The effect of the gut–brain axis before and after AUD/PTSD. Multiple sources establish that the gut–brain axis is causally linked to the pathophysiology of AUD [172,173] and that the gut–brain axis may have the potential to discover pathological mechanisms as well as therapeutic targets [174]. However, only a few attempts have been made on AUD/PTSD patients. (5) Future research should concentrate on the effects of DNA methylation and epigenetic regulation to study how they are controlled and altered in AUD/PTSD and consider the incorporation of fecal microbial translocation and other novel genetic manipulation and gene therapy techniques. (6) More research on the use of traditional herbal medicine and dietary modifications is required to improve brain abnormalities directly through antioxidant activities and/or restoration of the gut–brain axis. In recent years, beneficial effects of regulating AUD have been demonstrated in studies with herbal medicine and dietary supplements. In the future, the use of herbal medicine and dietary modifications may lead to new fields of prevention and therapeutic purposes [175,176]. (7) From the available psychopharmacology view, topiramate is a sensible choice for AUD, since it exhibits multiple pharmacologic effects [177]; although, additional effective drugs are encouraged to be found. (8) Finally, the researchers should pay attention to the co-occurrence of AUD/PTSD and conduct further investigations on this topic. For basic research and clinical applications, researchers are encouraged to screen and objectively identify state markers and biomarkers, as reported [178]. For instance, future research should include a procedure to identify objective forensic markers for clinically accurate diagnosis and prognosis evaluation. We hope that these suggestions can provide valuable information for future research.

## 7. Conclusions

Although many molecular mechanisms that may lead to AUD/PTSD comorbidity have been studied, the toxicological and pathophysiological mechanisms of AUD/PTSD remain largely unknown. Our review focuses on the molecular toxicology and pathophysiology of AUD and PTSD comorbidity, with significant additive or synergistic effects on the individual symptoms, compared with those suffering from each disease alone. We also emphasize that the patients with AUD/PTSD should be thoroughly examined and that the pathogenesis should be discussed in multiple aspects instead of under separate testing and consideration. We specifically discussed the molecular mechanisms of AUD, PTSD, and AUD/PTSD from metabolomics, including changes in molecular components in the brain, signal transduction pathways, gene expression, and epigenetics, and suggested possible hypotheses for the development of AUD/PTSD. This review focuses on certain aspects of molecular toxicology and pathophysiology mechanisms. Further studies on the causal relationship and mutual influence among AUD, PTSD, and AUD/PTSD are encouraged. 

## Figures and Tables

**Figure 1 ijms-24-08805-f001:**
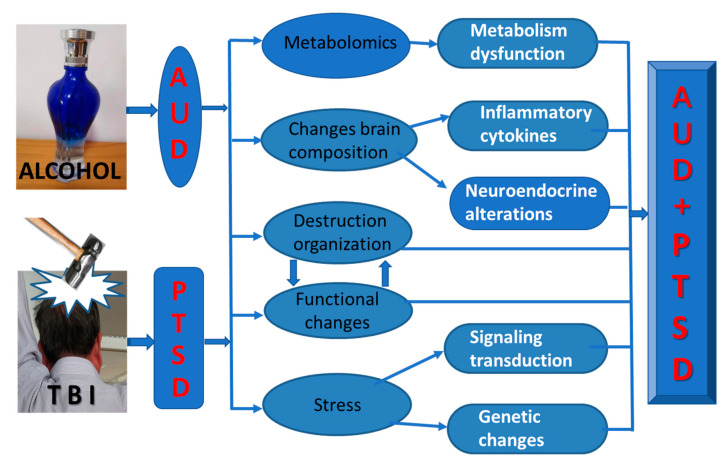
The molecular mechanisms of comorbid AUD/PTSD. AUD and PTSD following TBI result in abnormal metabolomics and metabolic dysfunction; alter the brain response resulting in abnormalities of inflammatory cytokines and neuroendocrine function; trigger stress responses in the body leading to changes in signal transduction systems, genes, and epigenetics; harm the structure and function of the brain; create an interconnected effect. Eventually, the combination of all risk factors causes AUD/PTSD.

**Table 1 ijms-24-08805-t001:** The diagnostic criteria of AUD and PTSD and the characteristics of AUD/PTSD.

Item	Diagnostic Criteria or Characteristics
The diagnostic criteria of AUD (DSM-5)	The DSM-5 defines AUD as a problematic pattern of alcohol use leading to clinically significant impairment or distress, as manifested by at least 2 of the following 11 symptoms occurring within 12-months [38,39,40]. (1) Alcohol is often taken in larger amounts or over a longer period than it is intended. (2) There is a persistent desire, or there are unsuccessful efforts, to reduce or control alcohol use. (3) A great deal of time is spent in activities necessary to obtain alcohol, use alcohol, or recover from its effects. (4) There is a craving, strong desire, or urge to drink alcohol. (5) Recurrent alcohol use results in a failure to fulfill major role obligations at work, school, or home. (6) There is continued alcohol use despite having persistent or recurrent social or interpersonal problems caused or exacerbated by the effects of alcohol. (7) Important social, occupational, or recreational activities are given up or reduced because of alcohol use. (8) There is recurrent alcohol use in situations where it is physically hazardous. (9) Alcohol use is continued despite knowledge of having a persistent or recurrent physical or psychological problem that is likely to have been caused or exacerbated by alcohol drinking. (10) There is tolerance, as defined by either of the following: a. There is a need for markedly increased amounts of alcohol to achieve intoxication or desired effect. b. There is a markedly diminished effect with continued use of the same amount of alcohol. (11) There is withdrawal, as manifested by either of the following: a. There are characteristic withdrawal syndromes for alcohol (the “How is alcohol withdrawal managed?” section for some DSM-5 symptoms of withdrawal). b. Alcohol (or a closely related substance, such as a benzodiazepine) is taken to relieve or avoid withdrawal symptoms.
The diagnostic criteria of PTSD (DSM-5)	Under DSM-5, for those older than six years of age, PTSD includes four clusters of symptoms [41]: (1) Re-experiencing the event: recurrent memories of the event, traumatic nightmares, dissociative reactions, prolonged psychological distress. (2) Alterations in arousal: aggressive, reckless, or self-destructive behavior; sleep disturbances; hypervigilance. (3) Avoidance: distressing memories, thoughts, or reminders of the event. (4) Negative alterations in cognition and mood: persistent negative beliefs, distorted blame, or trauma-related emotions; feelings of alienation and diminished interest in life.
The characteristics of AUD/PTSD	The main characteristics of comorbidity between PTSD and AUD following TBI: (1) Significant additivity of clinical symptoms with multiple indications of the two diseases. (2) Severe clinical hazards produce more serious clinical manifestations than any single disease in a vicious cycle. (3) Synergistic effect on the interaction and pathogenesis of AUD/PTSD. (4) With very little objective identification of state markers, the effectiveness of diagnosis and treatment is unsatisfactory.

DSM-5: Diagnostic and Statistical Manual of Mental Disorders, Fifth Edition; AUD: Alcohol Use Disorder; PTSD: Post-Traumatic Stress Disorder; TBI: Traumatic Brain Injury.

**Table 2 ijms-24-08805-t002:** Molecular component changes due to AUD, PTSD, and AUD/PTSD.

Change Composition	Effect	State	Reference
Metabolism/Pharmacokinetics			
Catalase	Ethanol oxidation in cortical brain tissue↑	PTSD	[56]
CYP2E1	Oxidative alcohol and acetaldehyde metabolism↑	AUD, AUD/PTSD	[57]
Glutamate	Forebrain reward circuit; Forebrain glutamate↑	AUD, AUD/PTSD	[58]
Cholesterol	Synaptogenesis, and synaptic communication↓	AUD	[59]
Phosphatidylethanol	Brain lipid membranes	AUD	[60]
*alpha* 6-nicotinic acetylcholine	Target sensitivity↓	AUD	[61]
Inflammatory factors			
CRP	CRP levels in the clinically elevated range↑	PTSD, AUD/PTSD	[62]
TLR4	Microglia activation/neuroinflammation↑	AUD	[63]
IL-6	Immunosuppressive effects↑	PTSD	[64]
Neuroendocrine components			
5-HT	Receptor responsiveness↑ Dopamine transmission↑	AUD	[65]
Serotonin transporter	Moderate associations between TBI and PTSD	PTSD	[66]
BDNF Va*l66Met*	Alcohol dependence	AUD	[67]
BDNF Met/Met genotype	Post-deployment PTSD score changes	PTSD	[68]
IGF-1	Plasma concentrations↓	AUD/PTSD	[69]
Gut-brain peptides	Regulate addictive behaviors	AUD/PTSD	[70]

PTSD: Post-Traumatic Stress Disorder; CYP2E1: Cytochrome P450-2E1; AUD: Alcohol Use Disorder; CRP: C-Reactive Protein; TLR4: Toll-Like Receptor 4; IL-6: Interleukin-6; 5-HT: 5-HydroxyTryptamine, Serotonin; TBI: Traumatic Brain Injury; BDNF: Brain-Derived Neurotrophic Factor; IGF-1: Insulin-like Growth Factor-1; ↑: Up regulation; ↓: Down regulation.

**Table 3 ijms-24-08805-t003:** Summary of genetic changes in the brain for AUD, PTSD, and AUD/PTSD.

Genes	Cause	State	Reference
DNA			
MCP-1	Methylation changes	Chronic AUD	[143]
CYP2E1	Gene expression changes	Chronic AUD hippocampus ↑	[144]
SNP changes	Epigenetic changes	PTSD	[62]
Polygenic	Induced gene changes	PTSD	[145]
Methylation	Gene changes	Higher risk PTSD	[143]
RNA			
lncRNAs	Synaptic plasticity	Adolescence	[146]
microRNAs	Neuroadaptations	Chronic AUD	[147]
miR-130a	miRNA expression alters	Prefrontal cortex AUD	[147]
miR-142	Neuroinflammation ↓	Alleviate PTSD	[148]
Chromosome			
11q14.2	Brain structural variation	AUD behavior	[149]
rs3852144	Emotional memory formation	PTSD development	[150]
Transcriptomics and genetics			
Atf3, c-Fos, Egr1, Npas4	Reduced downregulation	AUD after TBI	[151]
GRIN3B	mRNA level	Biomarker of PTSD	[152]
Gadd45b, Gadd45c	Upregulation	TBI	[151]
Parental AUD	More likely offspring AUD	AUD	[153]

MCP-1: Monocyte Chemotactic Protein 1; AUD: Alcohol Use Disorder; CYP2E1: Cytochrome P450-2E1; SNP: Single Nucleotide Polymorphism; PTSD: Post-Traumatic Stress Disorder; TBI: Traumatic Brain Injury; ↑: Up regulation; ↓: Down regulation. AUD: Alcohol Use Disorder; PTSD: Post-Traumatic Stress Disorder; TBI: Traumatic Brain Injury.

**Table 4 ijms-24-08805-t004:** Summary of important data that emerged from the review.

Subcategories	Characteristics of Participants or Animals	Assessments and Tests	Main DATA Findings	Reference
Epidemiological data	A representative US non-institutionalized civilian adult (≥18 years) sample (N = 36,309) as the 2012-2013 NESARC-III.	DSM-5 for AUD	AUD prevalence of 12-months and lifetime were 13.9% and 29.1%, respectively.	[12]
	A national sample of US adults (N = 2953) recruited from an online panel.	DSM-5 for PTSD	PTSD prevalence of lifetime, past 12-months, and past 6-months was 8.3%, 4.7%, and 3.8%, respectively.	[29]
	A stratified sample of 10,641 participants as part of the Australian national survey of mental health and wellbeing.	The composite international diagnostic interview	AUD is responsible for 24.1% of substance-use problems in PTSD. (DSM-substance-use disorders and ICD-10 personality disorders).	[30]
Metabolism data	Rat cortical contusion model.	The activity of catalase	Three-fold increase in catalase activities in a time course.	[56]
	The enzymes of ethanol oxidation in the homogenates from the perfused brains of rats and mice.	Gas chromatography	About 60% of the ethanol oxidation process in rodent brains may be attributed to catalase.	[83]
	Severe TBI in rats.	Microdialysis determination	Extracellular glutamate levels increased 9-fold compared with uninjured control rats.	[97]
Inflammatory factor data	A total of 2600 warzone-deployed marines.	Plasma CRP concentration	Each 10-fold increment in CRP concentration was associated with an odds ratio of a nonzero outcome (presence vs. absence of any PTSD symptoms) of 1.51- and 1-fold increase in outcome with a nonzero value (extent of symptoms when present) of 1.06.	[102]
	C57BL/6 mice and mice deficient in MCP-1 exposed to ethanol.	MCP-1 levels	Ethanol-induced microglial activation, neuroinflammation, and a drastic increase in the mRNA and protein levels of MCP-1.	[63]
	Patients with an injury severity score and abbreviated injury scale of the head of at least 3 were included upon arrival in the emergency room and grouped according to positive BAC (>0.5%, BAC) vs. less than 0.5% alcohol (no BAC).	Systemic IL-6 levels	Systemic IL-6 levels and leukocyte counts (IL-6: 65.0 ± 8.0 vs. 151.8 ± 22.3; leukocytes: 10.2 ± 0.9 vs. 13.2 ± 0.8, both *p* < 0.05) were significantly lower in BAC-positive patients.	[64]
**Neuroendocrine component** **data**	A total of 635 African-American substance-dependent men were recruited.	5-HT levels,5-HT transporter gene	The HTR3B Ser129 allele and low 5-HTTLPR activity had an additive effect on alcohol + drug dependence (OR = 6.0 (2.1–16.6)) that accounted for 13% of the variance.	[65]
	Both pre- and post-deployment data on 231 of 458 soldiers were analyzed.	BDNF Met/Met genotype	The BDNF Met/Met genotype accounted for 22% of the variance of post-deployment PTSD scores (R (2) = 0.22, *p* < 0.001).	[68]
	Abstinent AUD patients (N = 91) and healthy control subjects (N = 55).	Plasma concentrations of BDNF, IGF-1, and IGFBP-3	AUD patients displayed a high prevalence of dual diagnosis (39.3%) and comorbid substance-use disorders (40.7%). Plasma BDNF and IGF-1 concentrations were significantly lower in the alcohol group than in the control group (*p* < 0.001).	[69]
**Genetic data**	Genome-wide miRNA and mRNA expression were examined in postmortem PFCs of 23 European and Australian AUD cases and 23 matched controls using the Illumina HumanHT-12 v4 Expression BeadChip array.	Target gene prediction, gene set enrichment analysis, and DAVID functional annotation clustering analysis	Two miRNAs and 787 coding genes were differentially expressed in the PFC of AUD cases. Downregulation of miR-130a may lead to altered expression of a number of genes in the PFC of AUD.	[147]
	A total of 924 Northern Ugandan rebel war survivors; identified seven suggestively significant SNPs; *p* ≤ 1 × 10^−5^ for lifetime PTSD risk.	Genome-wide association studies	Emotional memory formation seems to decline with the increasing number of rs3852144 G-alleles, rendering individuals more resilient to PTSD development.	[150]
	Danish nationwide registers; 15,477 offspring with parental AUD and 154,392 reference individuals from the general population.	Parental AUD was defined as registration for AUD treatment. AUD in offspring was identified from medical, pharmacy, treatment, and cause of death registers.	Paternal AUD plus other mental disorders (hazard ratio (HR) = 2.27, 95% confidence interval (CI): 2.10–2.46) and paternal AUD alone (HR = 2.21, 95% CI: 2.07–2.36) were associated with higher offspring AUD risk.	[153]

NESARC-III: National Epidemiologic Survey on Alcohol and Related Conditions III; DSM: Diagnostic and Statistical Manual of Mental Disorders; AUD: Alcohol Use Disorder; PTSD: Post-Traumatic Stress Disorder; ICD: International Classification of Diseases; TBI: Traumatic Brain Injury; CRP: C-Reactive Protein; MCP-1: Monocyte Chemotactic Protein 1; IL-6: Interleukin-6; BAC: blood alcohol concentration (BAC); 5-HT: 5-HT: 5-HydroxyTryptamine, Serotonin; BDNF: Brain-Derived Neurotrophic Factor; IGF-1: Insulin-like Growth Factor-1; IGFBP-3: IGF-1 Binding Protein-3; PFC: prefrontal cortex; SNP: single nucleotide polymorphism; HR: hazard ratio; CI: Confidence Interval.

## Data Availability

Not applicable.

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
