# Peer review of "Molecular Toxicology and Pathophysiology of Comorbid Alcohol Use Disorder and Post-Traumatic Stress Disorder Associated with Traumatic Brain Injury"

_ijms, 2023, doi:10.3390/ijms24108805_

Round 1

Reviewer 1 Report

This is a very thorough and well-presented theoretical review of potential state markers for co-morbid AUD/PTSD.  In all honesty, I found this review very refreshing, there were no overt language concerns that I could detect and I found the article very educational.  There are a few placed in the text when a different font was employed that should be corrected prior to publication, but otherwise, I have no concerns with the submitted version of this article. 

Author Response

We are delighted that reviewer 1 considers our manuscript to be a comprehensive and well-presented theoretical review of potential state markers for co-morbid AUD/PTSD and our article is highly educational and has no concerns regarding language. With regards to the few instances where a different font was used in the text, we rectified them in this revised version. 

April 14, 2023

Reference: Manuscript ID: ijms-2283851

Title: “Molecular Toxicology and Pathophysiology of Co-morbid Alcohol Use Disorder and Post-Traumatic Stress Disorder”

Dear Editor,   

We are pleased that our manuscript has been reviewed by experts in the field. We are thankful that the Academic Editor comments the manuscript is well written and interesting and gives us the opportunity to revise our manuscript. We greatly graft the three reviewers and your comments to improve our manuscript and are delighted by the overall enthusiasm.

Please find below our responses to the questions and comments provided by the reviewers on a point-by-point basis (the changes are labeled with blue words). We believe that this revised manuscript has been substantially improved. After addressing each question, we hope that our revised manuscript merits be published in the International Journal of Molecular Sciences.

Please do not hesitate to contact me if you require any additional information.

Sincerely Yours, 

Xin Wang, Ph.D.

Director of Neuroapoptosis Drug Discovery Laboratory

Lead Investigator
Department of Neurosurgery

Brigham and Women’s Hospital, Harvard Medical School

Reviewer 2 Report

Dear Authors,

I have read your paper with interest and althouh I find it very interesting there are some details requiring improvement in my opinion:

1. The manuscript would benefit significantly with more epidemiologic detials. What is the prevalence of TBI in AUD population? what is prevalence of AUD in PTSD population? What percentage of TBI is a result of alcohol misuse? That data would show  the importance of AUD/PTSD relation.

2. I believe that adding a table with AUD and PTSD diagnostic criteria would be more useful for understanding the paper.

3. Table 1 - it is unclear for me. THe biochemical changes described in that Table - are they present in  AUD or TBI or PTSD or combination of those entitites? Please, clarify it.

4. PTSD is frequently (in this manuscript) presented as a neurobiological consequence of TBI. And that is a simplification. PTSD may develop in subjects who did not suffer from TBI and otherwise not all TBI victims suffer from PTSD. And I find that frequently in the bodu of the text consequences of TBI are presented as mechanisms of PTSD. THerefore I suggest to make a decision whether the manuscript is on AUD/PTSD relation or AUD/TBI relation or maybe AUD and PTSD in TBI survivors only.

5. THe role of glutamate transmission in AUD is well and briefly described. I would be very satisfied with similar description of role of glutamate in TBI and in development of PTSD.

6. Term "anterior lobe" was used which is confusing for me. Was it frontal lobe?

Author Response

Author's Reply to the Review Report (Reviewer 2)

Dear Authors,

I have read your paper with interest and althouh I find it very interesting there are some details requiring improvement in my o

pinion:

  1. The manuscript would benefit significantly with more epidemiologic detials. What is the prevalence of TBI in AUD population? what is prevalence of AUD in PTSD population? What percentage of TBI is a result of alcohol misuse? That data would show  the importance of AUD/PTSD relation.

Response: We appreciate very much Reviewer 2's interest in our paper. In response to the epidemiological question, we have conducted a thorough literature review and included the added texts and references to present the relevant epidemiological data (Page 3).

  1. I believe that adding a table with AUD and PTSD diagnostic criteria would be more useful for understanding the paper.

Response: We have added a new table 2 with AUD and PTSD diagnostic criteria and the characteristics of AUD and PTSD (Pages 5-6).

  1. Table 1 - it is unclear for me. THe biochemical changes described in that Table - are they present in  AUD or TBI or PTSD or combination of those entitites? Please, clarify it.

Response: We have added a column of state to Table 2 in the revised manuscript (previous Table 1) to more clearly indicate the biochemical changes described in AUD or PTSD, AUD/PTSD.

  1. PTSD is frequently (in this manuscript) presented as a neurobiological consequence of TBI. And that is a simplification. PTSD may develop in subjects who did not suffer from TBI and otherwise not all TBI victims suffer from PTSD. And I find that frequently in the bodu of the text consequences of TBI are presented as mechanisms of PTSD. THerefore I suggest to make a decision whether the manuscript is on AUD/PTSD relation or AUD/TBI relation or maybe AUD and PTSD in TBI survivors only.

Response: One focus of our manuscript is comorbid AUD and PTSD following TBI. As per the reviewer's concerns, in order to present this topic more clearly, we highlight that PTSD is the result of multiple damaging factors, among these factors; TBI is an important risk factor for the occurrence of PTSD, although not all TBI victims suffer from PTSD (Page 3).

  1. THe role of glutamate transmission in AUD is well and briefly described. I would be very satisfied with similar description of role of glutamate in TBI and in development of PTSD.

Response: We have added a similar description of the role of glutamate in TBI and the development of PTSD in the revised manuscript (Page 9).

  1. Term "anterior lobe" was used which is confusing for me. Was it frontal lobe?

Response: Thank you for your careful review. We have changed "anterior lobe" into “frontal lobe” (Page 14).

Dear Editor,   

We are pleased that our manuscript has been reviewed by experts in the field. We are thankful that the Academic Editor comments the manuscript is well written and interesting and gives us the opportunity to revise our manuscript. We greatly graft the three reviewers and your comments to improve our manuscript and are delighted by the overall enthusiasm.

Please find below our responses to the questions and comments provided by the reviewers on a point-by-point basis (the changes are labeled with blue words). We believe that this revised manuscript has been substantially improved. After addressing each question, we hope that our revised manuscript merits be published in the International Journal of Molecular Sciences.

Please do not hesitate to contact me if you require any additional information.

Sincerely Yours, 

Xin Wang, Ph.D.

Director of Neuroapoptosis Drug Discovery Laboratory

Lead Investigator
Department of Neurosurgery

Brigham and Women’s Hospital, Harvard Medical School

Reviewer 3 Report

Please see the uploaded document

Author Response

Author's Reply to the Review Report (Reviewer 3)

Overall, I found this review timely, original, well-conducted and scientifically sound. However, I have some suggestions aimed at improving the quality of the paper, and these are outlined below:

  1. In the introduction, a brief note on the fact that AUD and PTSD might be multidimensional, including several variants with different psychological and neurobiological underpinnings, should be added with appropriate references (please see and refer to following dois: 2174/1389450116666150506114108 and 10.1080/13651501.2019.1699575).

Response: We appreciate reviewer 3’s overall enthusiasm. We cited these two articles in the introduction (Page 2, References 5 and 6). The two important pieces of research provide a systematic introduction to PTSD and in-depth discussions of its hazards and related treatments.

  1. The paper has very interesting tables and one very informative image, but it lacks a Table reporting some interesting data that emerged from the review. For example, the most relevant articles should be included, critically evaluated and briefly explained in one Table.

Response: Based on the important comments, we have additionally provided a new Table 4 to report some interesting data that emerged from the review (Pages 16-17).       

  1. I guess why did the Authors not conduct a systematic review, and what can they tell us about this point? Perhaps this is a minor review limitation, but, in my opinion, this should be acknowledged.

Response: We are grateful for reviewer 3’s consideration. Indeed, we have not conducted a systematic review based on this minor review limitation. In another manuscript or work, we will conduct a systematic review.

  1. However, even if narrative and interesting, I would suggest the Authors introduce a section on Materials and Methods of the review to let us know how they conducted the literature searches with relevant key terms.

Response: We have added the suggested Materials and Methods section (Page 4).

  1. Even if not directly related to the review aims, but how (and if any) the available psychopharmacology might influence the course of AUD/PTSD regarding their potential action on reviewed variables?

Response: We have added the notion to the revised manuscript accordingly (Page 21).

  1. Translating into “real world” clinical practice and medicine, what possible clinical shreds of evidence might arise from the present review and what the Researchers do suggest to improve practice? Please add a brief résumé paragraph on “recommendations” in terms of integrative care.

Response: We have added the suggested contents and references to the revised manuscript accordingly (Page 21).

Dear Editor,   

We are pleased that our manuscript has been reviewed by experts in the field. We are thankful that the Academic Editor comments the manuscript is well written and interesting and gives us the opportunity to revise our manuscript. We greatly graft the three reviewers and your comments to improve our manuscript and are delighted by the overall enthusiasm.

Please find below our responses to the questions and comments provided by the reviewers on a point-by-point basis (the changes are labeled with blue words). We believe that this revised manuscript has been substantially improved. After addressing each question, we hope that our revised manuscript merits be published in the International Journal of Molecular Sciences.

Please do not hesitate to contact me if you require any additional information.

Sincerely Yours, 

Xin Wang, Ph.D.

Director of Neuroapoptosis Drug Discovery Laboratory

Lead Investigator
Department of Neurosurgery

Brigham and Women’s Hospital, Harvard Medical School

Reviewer 4 Report

In the present narrative review the Authors aimed to summarize the main characteristics of comorbidity between AUD and PTSD (AUD/PTSD) and tried to highlight the significance of a comprehensive understanding of the molecular toxicology and pathophysiological mechanisms of AUD/PTSD, particularly following traumatic brain injury, with a focus on the role of metabolomics, inflammation, neuroendocrine, signal transduction pathways, and genetic regulation.

Overall, I found this review timely, original, well-conducted and scientifically sound. However, I have some suggestions aimed at improving the quality of the paper, and these are outlined below:

  1. In the introduction, a brief note on the fact that AUD and PTSD might be multidimensional, including several variants with different psychological and neurobiological underpinnings, should be added with appropriate references (please see and refer to following dois: 10.2174/1389450116666150506114108 and 10.1080/13651501.2019.1699575).
  2. The paper has very interesting tables and one very informative image, but it lacks a Table reporting some interesting data that emerged from the review. For example, the most relevant articles should be included, critically evaluated and briefly explained in one Table.
  3. I guess why did the Authors not conduct a systematic review, and what can they tell us about this point? Perhaps this is a minor review limitation, but, in my opinion, this should be acknowledged.
  4. However, even if narrative and interesting, I would suggest the Authors introduce a section on Materials and Methods of the review to let us know how they conducted the literature searches with relevant key terms.
  5. Even if not directly related to the review aims, but how (and if any) the available psychopharmacology might influence the course of AUD/PTSD regarding their potential action on reviewed variables?
  6. Translating into “real world” clinical practice and medicine, what possible clinical shreds of evidence might arise from the present review and what the Researchers do suggest to improve practice? Please add a brief résumé paragraph on “recommendations” in terms of integrative care.

Author Response

Author's Reply to the Review Report (Reviewer 4)

Recent evidence indicates that the US are facing a public health crisis of alcohol misuse which has been partly carried out by significant rises in binge and heavy drinking behaviours. Moreover, this trend turned out to be significantly raised following the Covid-pandemic, and especially by the increasing prevalence of alcohol-related problems occurring in psychiatric patients. On this topic, the manuscript “Molecular Toxicology and Pathophysiology of Co­morbid Alcohol Use Disorder and Post-Traumatic Stress Disorder” represents an interesting exploration in the attempt to cope with such a health issue.

The paper appears to suffer from some major issues that need extensive revision. First, despite the proposal of the authors appears to be sufficiently adequate, the thesis statement is not clearly and accurately reported since the title of the manuscript, and the introduction further seems to imply some difficulties in the overall understanding of the main topic in terms of coherence and cohesion.

  1. Namely, it is not clear how you come to introduce the TBI topic. What exactly is TBI? Why is it important to assess it? What is its incidence? What is the relationship with AUD? and with PTSD? Most importantly, why do the authors consider it relevant to describe AUD/PTSD co-occurrence from the perspective of TBI -moreover, citing as primary sources previous works which do not deal primarily with TBI. For the overall understanding of this paper, this is also relevant because TBI is considered by the authors to be a key factor for the entire description and final suggestion.

Thus, any potential causal relationship between TBI and AUD or PTSD is much more

complex than that described by the authors. Indeed, the opposite relationship is also

true, in that alcohol use might be the consequence of TBI, but TBI is also one of the

most important acute consequences of alcohol use.

Therefore, the authors are invited to consider further review of this relationship. And

a more extensive review of the literature is recommended -i.e.: Weil ZM, Corrigan JD,

Karelina K. Alcohol Use Disorder and Traumatic Brain Injury. Alcohol Res.

2018;39(2):171-180. PMID: 31198656; PMCID: PMC6561403.

Also, in this regard, perhaps the title of the manuscript should be changed to refer to

TBI, and a section on limitations should be added at the end of this paper.

Response: Thanks for reviewer 4's positive comments that our manuscript represents an interesting exploration in the attempt to cope with the important current health issue. Based on the constructive criticism and to more clearly and accurately express the main topic, we have added the words “associated with TBI” in the title and included significant TBI relevant contents in the Introduction of the revised manuscript (Pages 3-4). In addition, we have cited the important review article entitled “Alcohol Use Disorder and Traumatic Brain Injury”. Alcohol Res. PMID: 31198656 (Page 3). Furthermore, in terms of coherence and cohesion, at the end of this paper, we have added related prospects for the mutual influence and complex relationships among TBI, PTSD, and AUD.

  1. As a second point, it is not clear which are the most relevant markers of AUD/PTSD co-occurrence status. Since the main focus of this paper is "identification of co­morbidity state markers," and the description and tables report observations mainly of AUD or PTSD separately, this topic seems of primary importance for the coherence of the entire manuscript, and therefore requires more attention.

Response: At present, there are no recognized co­morbidity state markers for AUD/PTSD cooccurrence, and research on the identification of markers mainly focuses on individual AUD or PTSD. Although there are some markers for AUD, there are no objectively recognized markers for PTSD. Thus, this is a challenging topic and there are no markers for PTSD and AUD/PTSD cooccurrence. The current strategy is to conduct screening and validation on existing preliminary results of markers, with the purpose of ultimately discovering the identification markers. At the end of this article, we propose further research directions accordingly.

  1. Third, many theories have been provided to try to understand the relationship between AUD and stress-related disorders, such as PTSD. To name the most popular, the self-medication hypothesis proposed by Khantzian; an alcoholic lifestyle that increases the risk for patients with AUD to incur trauma, then to develop PTSD; a common underlying vulnerability (based on genetics, for example). In addition, a description of the neuro-endocrine aspects of this topic must take into consideration the implication of the HPA axis -and cortisol -on both PTSD and AUD, and their comorbidity. Therefore, authors should consider extending their introduction with observations on these topics, and perhaps add it as a limitation.

Response: Based on these important suggestions, we have added “the self-medication hypothesis” and cited the paper of Khantzian in the revised manuscript (Page 21). Additionally, we have added/modified texts to show that an alcoholic lifestyle increases the risk for patients with AUD to incur trauma, then develop PTSD (Page 3). Furthermore, we have added a suggested description of the neuro-endocrine aspects of this topic to take into consideration the implication of the HPA axis and cortisol (Pages 12-14). Overall, we have extended the work on these topics.

Formal revision of this paper was not eased by the absence of line numbers on the side

of the paper.

Response: We have added the side line numbers in the revised manuscript.

A small revision of the format style in the introduction and tables is required.

Response: We have modified the format style in the introduction (Page 3) and Tables (Page 14).

The main suggestions are the following: Page 2, paragraph 1. ‘According to epidemiological studies on post-war veterans, have shown that those with PTSD were likely […]’ -where is the subject of the present statement?

Response: We have modified the mentioned texts (Page 3).

-Page 2, paragraph 2. ‘Alcohol (ethanol) is the most frequently used recreational agent in the world’ -Any references?

Response: We have added the new references [36] and [37] and modified the sentence in the revised manuscript (Page 4).

 -Page 3, paragraph 2. ‘As a result, the harm caused by the comorbidity of AUD and PTSD is no longer a medical issue but has evolved into an economic and social problem that is gaining attention’ – AUD and PTSD are both medical issues, the same applies to AUD/PTSD co-occurrence regardless of whether you consider it in its short-or long-term course. Please therefore consider adding a 'not only' before 'a medical issue'.

Response: We have added a ‘not only’ and changed ‘but’ into ‘but also’ on Page 5.

-Page 3, paragraph 3. ‘Understanding the synergistic interactions between AUD and PTSD, as well as their co-morbid molecular toxicology and pathophysiological mechanisms, have become critical for better prevention and treatment of AUD/PTSD symptoms’– please reconsider the subject-verb agreement.

Response: We modified this sentence (Page 7).

-Page 4, line 1. ‘Identification’ does not need the Uppercase letter;

Response: We have corrected it (Page 6).

-Page 5, lines 1-2. The statement ‘long-term harm of alcohol cytotoxicity is obvious in patients with long-term chronic misuse’ is not clear. Further explanation is needed.

Response: We have modified this sentence in the revised manuscript (Page 8).

-Page 5, line 8. Replace the comma with a period before "For example […]". Reword the following line, for example, by adding an "and" before "Salivary," which does not need to be capitalized in this case. Further, in the same paragraph, ‘[…] Inhibiting CYP2E1’ does not need the uppercase letter.

Response: We have modified the mentioned texts accordingly (Page 9).

-Second half of page 5. What is the BBB? Need to explain, or at least report, the entire name from which the acronym is derived.

Response: We have added the entire name Blood Brain Barrier (BBB) in the text (Page 9)

-Paragraph 3.2, line 6. Remove one full stop.

Response: We have corrected it (Page 10).

In the end, the final paragraph appears dense in content. It is suggested that it be split into two subsections, perhaps: ‘Implications and research […]’, and ‘Conclusion’.

Response: We are grateful for the suggestion and have split the final paragraph into two subsections: 6. Implications and further research directions (Pages 21-22) and 7. Conclusion (Pages 22-23).

Dear Editor,   

We are pleased that our manuscript has been reviewed by experts in the field. We are thankful that the Academic Editor comments the manuscript is well written and interesting and gives us the opportunity to revise our manuscript. We greatly graft the three reviewers and your comments to improve our manuscript and are delighted by the overall enthusiasm.

Please find below our responses to the questions and comments provided by the reviewers on a point-by-point basis (the changes are labeled with blue words). We believe that this revised manuscript has been substantially improved. After addressing each question, we hope that our revised manuscript merits be published in the International Journal of Molecular Sciences.

Please do not hesitate to contact me if you require any additional information.

Sincerely Yours, 

Xin Wang, Ph.D.

Director of Neuroapoptosis Drug Discovery Laboratory

Lead Investigator
Department of Neurosurgery

Brigham and Women’s Hospital, Harvard Medical School

Round 2

Reviewer 2 Report

Dear Authors,

THank You for addressing my concerns. I am satisfied and I do not have further questions.

Author Response

Thanks so much for reviewer 2's positive comments.

Reviewer 3 Report

The authors have sufficiently revised the original version of the manuscript. The current version of the article appears coherent and readable to a great extent, even though the section on neuroendocrine aspects needs further work. Namely, the relationships between HPA axis activity and both stress-related disorders (such as PTSD) and AUD require major attention in the section. Specifically, the references do not seem sufficiently coherent with the topic, and the sentence 'Chronic alcohol exposure enhances the HPA axis's tolerance to alcohol, resulting in lower blood cortisol levels after alcohol consumption in AUD patients compared to individuals who do not have alcohol use disorders' seems to be a strong statement, or at least requires to be properly contextualised.

To explain, over the past decade, dysregulation of the endogenous stress-regulation systems is shown to play a key role across several psychiatric conditions, such as PTSD (see Dunlop BW & Wong A 2019), and SUD (see Uhart M & Wand GS 2009, Zorrilla EP 2014). Decades of research on cortisol levels in chronic substance users have substantially shown higher basal cortisol levels compared with controls, as reported by studies mainly based on saliva or plasma measurements in chronic substance users, or binge/disordered alcohol drinkers (see Adinoff B 2003). For a more detailed description on this topic, see Wemm & Sinha’s review (Wemm SE & Sinha R 2019). Moreover, individual cortisol assessments are strongly influenced by the acute context of the measurement situation (time of day, day of week, specific circumstances) and may therefore provide rather poor intra-individual reliability and as a cross-sectional measure. Therefore, it is considered an inadequate reflection of normal long-term cortisol secretion. In contrast with short-term cortisol measurements, hair analysis offers the opportunity to measure the long-term activity of the HPA axis through a retrospective assessment of cumulative cortisol exposure over time (see Stalder T & Kirschbaum C 2012Stalder T 2017). Therefore, to date, hair cortisol analysis has considered to be the most reliable method for measuring HPA axis activity, both with research and clinical purpose.

Relevant results showing elevated HCC in chronic substance users such as alcohol, psychostimulants and opioids have also been provided. To assess the relationship between chronic alcohol use and long-term HPA axis activity, a few studies have been made. See Stalder et al. 2010 and Price JL & Nixon SJ 2021 for details.

Finally, interestingly, studies on psychiatric patients diagnosed with more than a single mental disorder - or on substance users with psychiatric disorders - have revealed that co-occurrence of psychiatric disorders might be an additional correlate of HPA axis dysregulation.

Author Response

Author's Reply to the Review Report (Reviewer 3)

Reviewer 3

Comments and Suggestions for Authors

The authors have sufficiently revised the original version of the manuscript. The current version of the article appears coherent and readable to a great extent, even though the section on neuroendocrine aspects needs further work. Namely, the relationships between HPA axis activity and both stress-related disorders (such as PTSD) and AUD require major attention in the section. Specifically, the references do not seem sufficiently coherent with the topic, and the sentence 'Chronic alcohol exposure enhances the HPA axis's tolerance to alcohol, resulting in lower blood cortisol levels after alcohol consumption in AUD patients compared to individuals who do not have alcohol use disorders' seems to be a strong statement, or at least requires to be properly contextualised.

To explain, over the past decade, dysregulation of the endogenous stress-regulation systems is shown to play a key role across several psychiatric conditions, such as PTSD (see Dunlop BW & Wong A 2019), and SUD (see Uhart M & Wand GS 2009, Zorrilla EP 2014). Decades of research on cortisol levels in chronic substance users have substantially shown higher basal cortisol levels compared with controls, as reported by studies mainly based on saliva or plasma measurements in chronic substance users, or binge/disordered alcohol drinkers (see Adinoff B 2003). For a more detailed description on this topic, see Wemm & Sinha’s review (Wemm SE & Sinha R 2019). Moreover, individual cortisol assessments are strongly influenced by the acute context of the measurement situation (time of day, day of week, specific circumstances) and may therefore provide rather poor intra-individual reliability and as a cross-sectional measure. Therefore, it is considered an inadequate reflection of normal long-term cortisol secretion. In contrast with short-term cortisol measurements, hair analysis offers the opportunity to measure the long-term activity of the HPA axis through a retrospective assessment of cumulative cortisol exposure over time (see Stalder T & Kirschbaum C 2012, Stalder T 2017). Therefore, to date, hair cortisol analysis has considered to be the most reliable method for measuring HPA axis activity, both with research and clinical purpose.

Response: We appreciate very much reviewer 3’s constructive criticism. We have added the significant recommended and new references to be sufficiently coherent with the topic. Moreover, we have provided more attention to the relationships between HPA axis activity and stress-related disorders PTSD and AUD and have the sentence 'Chronic alcohol exposure enhances the HPA axis's tolerance to alcohol, resulting in lower blood cortisol levels after alcohol consumption in AUD patients compared to individuals who do not have alcohol use disorders' to be properly contextualized. These new pieces of recommended researches additionally discuss the endogenous stress-regulation systems, HPA axis, chronic substance use disorder, short-term and long-term cortisol levels measurement, etc. These important contents greatly increase the credibility and persuasiveness of the article (Pages 11-13).

Relevant results showing elevated HCC in chronic substance users such as alcohol, psychostimulants and opioids have also been provided. To assess the relationship between chronic alcohol use and long-term HPA axis activity, a few studies have been made. See Stalder et al. 2010 and Price JL & Nixon SJ 2021 for details.

Response: Based on the important suggestion, we have added relevant texts (Pages 12-13). 

Finally, interestingly, studies on psychiatric patients diagnosed with more than a single mental disorder - or on substance users with psychiatric disorders - have revealed that the co-occurrence of psychiatric disorders might be an additional correlate of HPA axis dysregulation.

Response: We have added the relevant content accordingly (Page 22).